# Acute and long-term kidney function after parathyroidectomy for primary hyperparathyroidism

**Marcelo Belli** [1]*, **Regina Matsunaga Martin**[2], **Marília D'Elboux Guimarães Brescia**[1], **Climério Pereira Nascimento, Jr**[1], **Ledo Mazzei Massoni Neto**[1], **Sergio Samir Arap**[1], **Bruno Ferraz-de-Souza**[3], **Rosa Maria Affonso Moyses**[3], **Munro Peacock**[4], **Fábio Luiz de Menezes Montenegro**[1]

1 Division of Head and Neck Surgery, Hospital das Clinicas HCFMUSP, Faculdade de Medicina, Universidade de São Paulo, São Paulo, Brasil, 2 Divison of Endocrinology, Hospital das Clinicas HCFMUSP, Faculdade de Medicina, Universidade de São Paulo, São Paulo, Brasil, 3 Divison of Nephrology, Hospital das Clinicas HCFMUSP, Faculdade de Medicina, Universidade de São Paulo, São Paulo, Brasil, 4 Department of Medicine, Indiana University School of Medicine, Indianapolis, Indiana, United States of America

* marcelobelli33@gmail.com

**Data Availability Statement:** All relevant data are within the manuscript and its Supporting information files.

## Abstract

### Background

In kidney transplant patients, parathyroidectomy is associated with an acute decrease in renal function. Acute and chronic effects of parathyroidectomy on renal function have not been extensively studied in primary hyperparathyroidism (PHPT).

### Methods

This retrospective cohort study included 494 patients undergoing parathyroidectomy for PHPT. Acute renal changes were evaluated daily until day 4 post-parathyroidectomy and were stratified according to acute kidney injury (AKI) criteria. Biochemical assessment included serum creatinine, total and ionized calcium, parathyroid hormone (PTH), and 25-hydroxyvitamin D (25OHD). The estimated glomerular filtration rate (eGFR) was calculated using the CKD-EPI equation. We compared preoperative and postoperative renal function up to 5 years of follow-up.

### Results

A total of 391 (79.1%) patients were female, and 422 (85.4%) were non-African American. The median age was 58 years old. The median (first and third quartiles) preoperative serum creatinine, PTH and total calcium levels were 0.81 mg/dL (0.68–1.01), 154.5 pg/mL (106–238.5), and 10.9 mg/dL (10.3–11.5), respectively. The median (first and third quartiles) pre-operative eGFR was 86 mL/min/1.73 m$^2$ (65–101.3). After surgery, the median acute decrease in the eGFR was 21 mL/min/1.73 m$^2$ (p<0.0001). Acutely, 41.1% of patients developed stage 1 AKI, 5.9% developed stage 2 AKI, and 1.8% developed stage 3 AKI. The acute eGFR decrease (%) was correlated with age and PTH, calcium and preoperative

**Funding:** The author(s) received no specific funding for this work.

**Competing interests:** The authors have declared that no competing interests exist.

creatinine levels in univariate analysis. Multivariate analysis showed that the acute change was related to age and preoperative values of ionized calcium, phosphorus and creatinine. The change at 12 months was related to sex, preoperative creatinine and 25OHD. Permanent reduction in the eGFR occurred in 60.7% of patients after an acute episode.

## Conclusion

There was significant acute impairment in renal function after parathyroidectomy for PHPT, and almost half of the patients met the criteria for AKI. Significant eGFR recovery was observed during the first month after surgery, but a small permanent reduction may occur. Patients treated for PHPT seemed to present with prominent renal dysfunction compared to patients who underwent thyroidectomy.

## Introduction

Kidney function in renal transplant patients following parathyroidectomy for tertiary hyperparathyroidism has attracted considerable attention. According to some series of patients, the extent of parathyroidectomy affects long-term renal function [1, 2]. According to others, despite acute postoperative changes, most patients recover renal graft function with no significant long-term decrease compared with transplant patients undergoing other types of operations [3, 4]. The degree of renal function impairment after parathyroidectomy is correlated with serum parathyroid hormone (PTH) reduction [5]. PTH as well as PTH-related protein can stimulate glomerular filtration, glomerular blood flow and urine output independent of calcium action [6, 7]. Other factors altered in secondary hyperparathyroidism due to renal failure, such as fibroblast growth factor 23 (FGF-23) and the renin-angiotensin-aldosterone system, may also affect renal function after parathyroidectomy [8, 9].

A prospective study of 62 patients treated surgically for primary hyperparathyroidism (PHPT) reported acute renal impairment with a significant reduction in the estimated glomerular filtration rate (eGFR) that was sustained over 30 days of follow-up [10]. The authors attributed the acute elevation in serum creatinine to preoperative comorbidities. PTH was not reported, and creatinine was measured only once in the acute period. In a retrospective study of long-term kidney function after parathyroidectomy for PHPT, a permanent reduction in the eGFR occurred in patients with a preoperative eGFR $\geq$ 60 mL/min/1.73 m$^2$ [11]. Assuming that untreated PHPT progressively decreases renal function, the authors suggested that parathyroidectomy halted the deterioration of renal function in these patients with better kidney function prior to treatment.

Currently, there is no recommendation for renal function surveillance after parathyroidectomy in PHPT treatment guidelines, and no attention has been paid to the risk of acute kidney injury (AKI), even in a large series of patients treated surgically [12–14]. However, renal failure comprises 9.54% of the diagnoses as a cause of nonelective readmission after parathyroidectomy for PHPT [15]. In the present retrospective study of 494 surgically proven PHPT patients, we documented both the acute and chronic effects of parathyroidectomy on renal function and related these changes to serum biochemistry results before surgery.

## Materials and methods

We conducted this retrospective cohort study of patients surgically treated for PHPT at Hospital das Clinicas, an academic hospital in Sao Paulo, Brazil. The study was approved by Ethics Committee of Hospital das Clínicas da Faculdade de Medicina da Universidade de São Paulo (#53784/#469.905) and written informed consents were not required because the data were analyzed anonymously.

The period of study ranged from January 2007 to December 2016. Patients were included when parathyroidectomy was successful, as judged by a decrease in high serum calcium and PTH levels to reference ranges. The exclusion criteria included an age less than 18 years old, subjects with no measurement of renal function during the hospital stay (acute period) or after discharge (follow-up period), and patients with a basal eGFR <15 mL/min/1.73 m$^2$ (stage 5 chronic kidney disease).

The data collected included age at surgery, sex, ethnicity and type of parathyroid disease. Patients with uniglandular disease verified by intraoperative PTH assay underwent focalized surgery under general anesthesia. When the cause of PHPT was type 1 multiple endocrine neoplasia (MEN1) or sporadic hyperplasia, total parathyroidectomy with auto transplantation (TPTX + AT) in the forearm was mostly performed, but some patients underwent subtotal parathyroidectomy at the discretion of the operating team.

Preoperative and postoperative biochemical data were retrieved from a hospital electronic database. Serum creatinine was measured by kinetic colorimetric assay (normal range, 0.7 to 1.2 mg/dL for men and 0.5 to 0.9 mg/dL for women) as a biochemical marker of renal function (coefficient of variation [CV] of 1.9% for a concentration of 1.5 mg/dL). According to the hospital surgical protocol, all patients were hospitalized for at least three days postoperatively. Serum creatinine was measured preoperatively and daily after surgery and prior to ambulatory visits at 1, 3, 6, 12, 24, 36, 48 and 60 months. Preoperatively, biochemical assessment also included total and ionized calcium, phosphorus, PTH and 25-hydroxyvitamin D (25OHD). After surgery, serum total and ionized calcium were measured from day 1 to day 3. PTH (normal 10–65 pg/mL) was measured by electrochemiluminescence immunoassay, and 25OHD (sufficiency > 30 ng/mL) was measured by chemiluminescence immunoassay (Lyason-Diasorin analyzer with a CV of 8.1%).

The 25OH vitamin D replacement data were not controlled during the study period. Oral supplementation with cholecalciferol, when indicated, was performed according to the Brazilian Society of Endocrinology and Metabology recommendations [16].

The eGFR values were calculated from serum creatinine, age and ethnicity using an internet-based electronic calculator of the Chronic Kidney Disease Epidemiology Collaboration (CKD-EPI) formula, available at [http://mdrd.com/] [17].

The proportion of patients meeting the criteria for AKI was calculated using the highest value for serum creatinine (Cr PO) measured until day 4. The Improving Global Outcomes (KDIGO) Acute Kidney Injury Work Group divides AKI into three stages according to changes in serum creatinine: stage 1 (increase of 0.3 mg/dL or 1.5 to 1.9 times baseline), stage 2 (2.0 to 2.9 times baseline) and stage 3 (>3.0 times baseline, or increase ≥ 4.0 mg/dL, or initiation of dialysis) [18, 19]. Patients not fulfilling the criteria for AKI constituted the "no AKI group". In a few cases, creatinine decreased acutely, and these cases were grouped as the "S+ group".

To test whether preoperative chronic kidney disease (CKD) was a factor influencing the biochemistry response, we used the GFR categories proposed by the KDIGO Clinical Practice Guideline for the Evaluation and Management of CKD [20, 21]. This guideline stratifies CKD into group 1 (G1), normal or elevated (eGFR ≥ 90 mL/min/1.73 m$^2$); group 2 (G2), mildly

decreased (eGFR of 89–60 mL/min/1.73 m$^2$); group 3 (G3), mildly to severely decreased (eGFR of 59–30 mL/min/1.73 m$^2$); group 4 (G4), severely decreased (eGFR of 29–15 mL/min/1.73 m$^2$); and group 5 (G5), kidney failure (eGFR < 15 mL/min/1.73 m$^2$) [11].

We analyzed whether the acute renal changes after parathyroidectomy were correlated with age, preoperative eGFR, PTH, phosphorus, serum calcium or type of renal failure.

An acute change in the eGFR (%) was calculated by the following formula: (acute postoperative eGFR–preoperative eGFR) X 100/preoperative eGFR. Accordingly, we obtained the 12-month postoperative eGFR chronic change in eGFR (%) and the acute change in serum calcium (%) by the following formulae: (12-month postoperative eGFR–preoperative eGFR) X 100/preoperative eGFR, and (day 1 postoperative calcium–preoperative calcium) X 100/preoperative calcium, respectively.

## Statistical analysis

Continuous variables were tested for normality with D'Agostino & Pearson and Kolmogorov-Smirnov normality tests. Variables with a normal distribution are presented as the mean and standard deviation (SD). Inferential statistics for these distributions included *t* tests and analysis of variance (ANOVA) for multiple comparisons. Whenever appropriate, repeated measures tests were used.

Nonparametric distributions are presented as medians and first and third quartiles (Q1-Q3). When at least one distribution was non-Gaussian, all calculations of inferential statistics were also nonparametric. Kruskal-Wallis and Dunn's multiple comparisons tests were employed for nonparametric unpaired continuous data. The Mann-Whitney test compared individual median ranks. The Friedman test was used for nonparametric paired data, and the Wilcoxon matched-pairs signed rank test compared 2 nonparametric paired group differences. Data were processed in GraphPad Prism® version 7.0a.

Acute and chronic renal changes were correlated with age, sex and preoperative biochemical values in bivariate analysis by Spearman's correlation coefficient. Multiple linear regression models were adjusted, including all variables with descriptive levels below 0.10 in bivariate analysis, using the stepwise forward method. We used IBM-SPSS for Windows version 20.0 for multivariate analysis.

## Results

### Overall cohort characteristics

From 2007 to 2016, 521 patients underwent parathyroidectomy for the treatment of PHPT, 27 of whom met the exclusion criteria. There were 3 deaths recorded in the perioperative period (30 days after parathyroidectomy) from a total sample of 521 patients, representing a 0.58% mortality rate. One patient died 6 days after surgery from sepsis; one died 14 days after surgery, along with new hospitalization caused by severe hypocalcemia and renal failure; and another died 20 days after surgery from an unknown cause.

After exclusion, 494 patients were included in the study. Among them, 391 (79.1%) were women, and 422 (85.4%) were non-African American. The median age (range) at surgery was 58 years old (18–86).

### Change in eGFR (%) in the PHPT group

Serum preoperative creatinine ranged from 0.28 to 3.76 mg/dL, with a median (Q1-Q3) of 0.81 mg/dL (0.68–1.01). During the hospital stay, serum creatinine measured daily until day 4 showed a reduction in 30 (6.1%) patients, remained unchanged in 6 (1.21%) and increased in

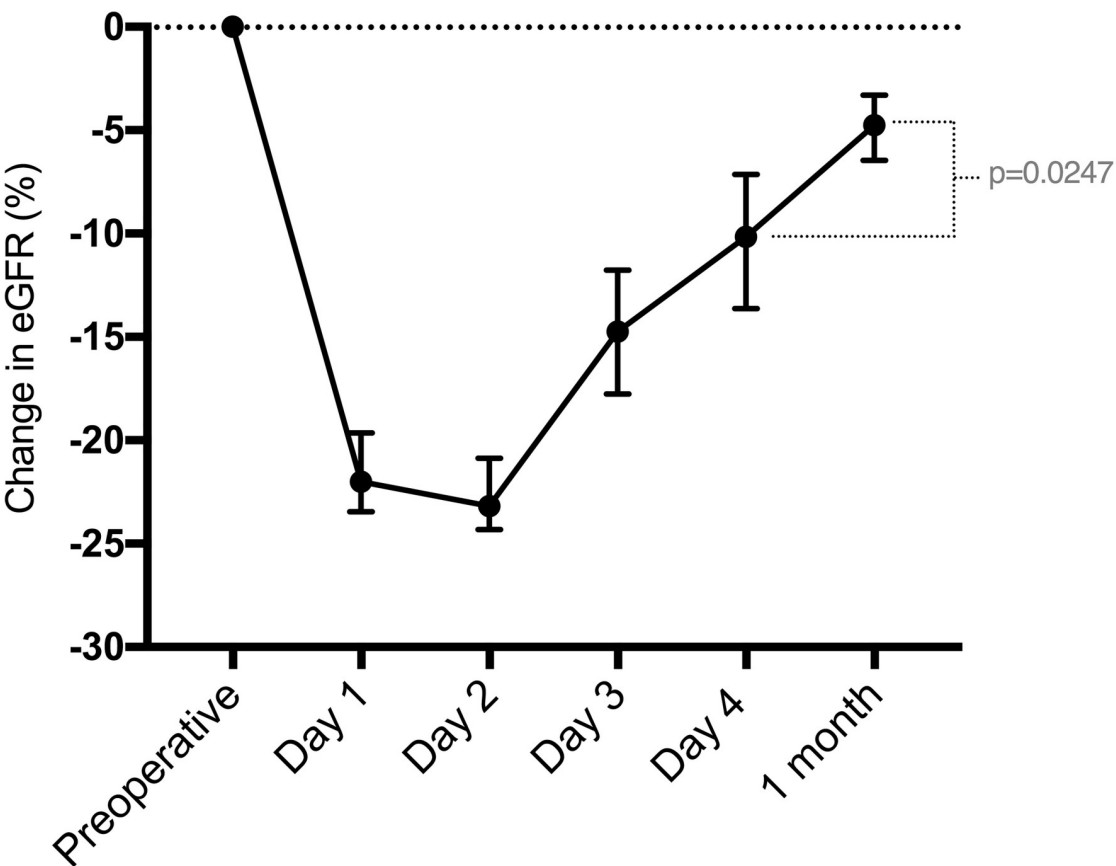

**Fig 1. Median and 95% confidence interval of the change in eGFR (%) compared to baseline, from the preoperative period until 1 month postoperation.**

458 (92.71%). Regarding the peak elevation of creatinine, the highest value of creatinine was found in 220 patients (44.5%) on day 1 after the operation, in 193 (39.1%) on day 2, in 54 (10.9%) on day 3 and in 27 (5.5%) on day 4.

In the acute period after parathyroidectomy, the baseline eGFR differed from the eGFR on days 1, 2, 3 and 4 (p<0.0001). Among the daily measurements, the median eGFR on day 1 was similar to that on day 2 (p = 0.8529) and day 3 (p = 0.1406) but was different from that on day 4 (p = 0.0281). The absolute eGFR change between day 4 and 1 month was also significant (p = 0.0247) (Fig 1). The median preoperative eGFR was significantly higher than at 1 month (p = 0.0006).

Compared to baseline, the median eGFR PO (acute) decrease was 21 mL/min/1.73 m$^2$ (p<0.0001). Over the first postoperative month, there was an eGFR recovery of 17 mL/min/ 1.73 m$^2$ (p<0.0001). The preoperative eGFR was significantly higher than all median eGFR values over 60 months postoperatively, and the 1-month eGFR was similar to all median values during follow-up (Table 1) (Fig 2).

In 284 patients selected for paired analysis (complete measures) of the baseline eGFR PO (acute) and at 1, 12 and 24 months postoperatively, the baseline eGFR differed from all eGFR values over 24 months (p<0.0001) (Fig 3). There was no significant difference between the 1-month eGFR and the eGFR at 12 (p = 0.9821) and 24 months (p = 0.9742).

**Table 1.  Estimated GFR (mL/min/1.73 m²) from preoperatively to 60 months postoperatively.**

|  | eGFR preop | eGFR PO ᵃ | eGFR 1 m | eGFR 6 m | eGFR 12 m | eGFR 24 m | eGFR 36 m | eGFR 48 m | eGFR 60 m |
|---|---|---|---|---|---|---|---|---|---|
| N | 494 | 494 | 395 | 409 | 450 | 362 | 323 | 255 | 193 |
| Min-Max | 16–150 | 10–135 | 14–150 | 5–138 | 7–148 | 12–136 | 13–145 | 10–137 | 8–139 |
| Median | 86 | 60 | 78 | 77 | 81.5 | 80 | 80 | 82 | 80 |
| Q1-Q3 | 65–101.3 | 41–80.3 | 59–96 | 56–95.5 | 59–96 | 60–97 | 61–97 | 61–97 | 63.5–96 |

ᵃ eGFR PO represents the lower eGFR values between day 1 and day 4 (acute period)

The mean acute change in the eGFR was -27.44% (SD ± 19.12%) and was correlated with age, PTH, and total and ionized calcium and phosphorus, as well as the preoperative eGFR (Table 2).

A chronic (permanent) decrease in the eGFR was observed in 60.7% of patients 12 months postoperation and represented a median (Q1-Q3) percentage difference of -4.42% (-15 to 4.48). Moreover, 5.8% of patients had the same eGFR value as that at baseline (preoperative), and 33.6% showed improvement in renal function 1 year after parathyroidectomy.

## Causes of PHPT

Most patients had sporadic PHPT with a single adenoma (351, 71.1%), and type 1 multiple endocrine neoplasia (MEN1) ranked second (91, 18.4%). The biochemical variables stratified by the cause of PHPT are shown in Table 3.

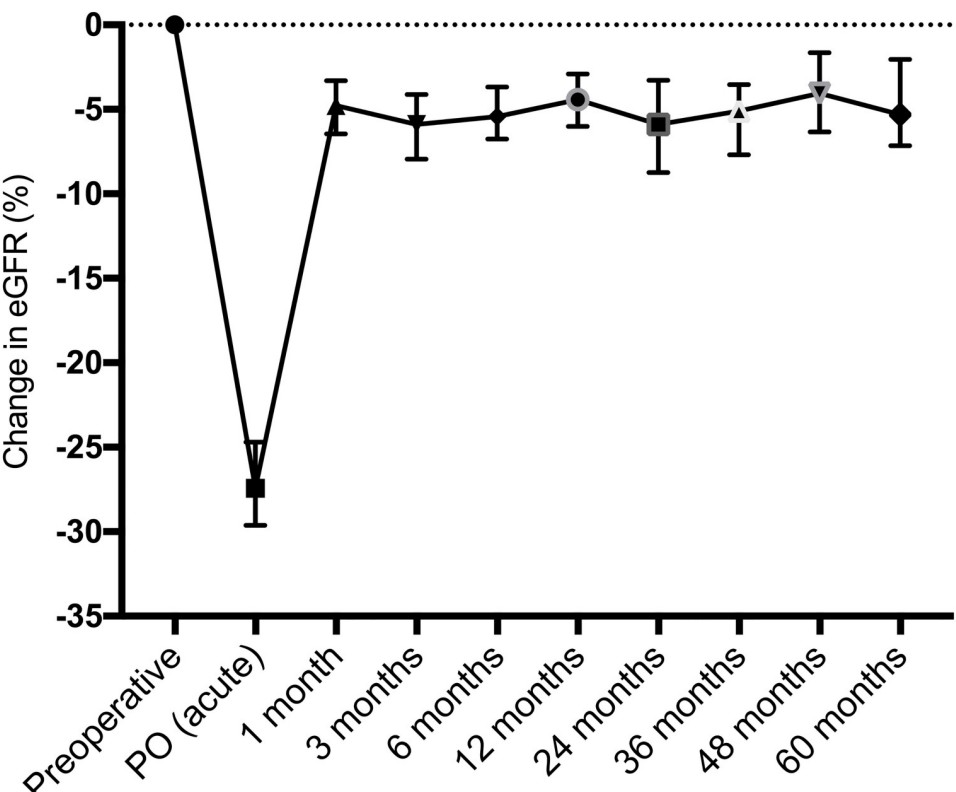

**Fig 2.  Median and 95% confidence interval of the change in eGFR (%) compared to baseline, from the preoperative period until 60 months of postoperative follow-up.**

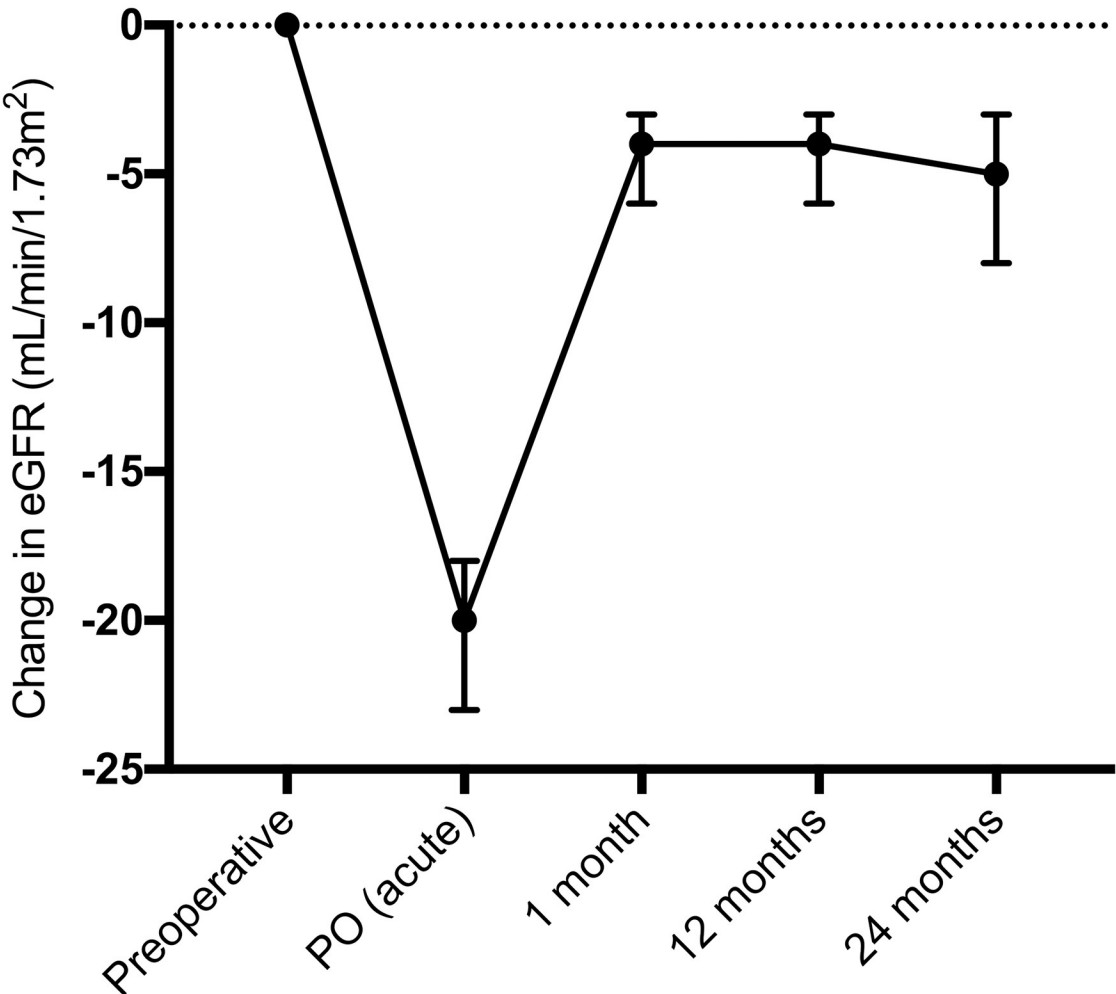

**Fig 3. Median and 95% confidence interval of the change in eGFR (mL/min/1.73 m$^2$) compared to baseline in patients with complete measures (n = 284) until 24 months postoperation.**

The preoperative eGFR was higher in MEN 1 patients than in the adenoma, double adenoma and sporadic hyperplasia groups (p<0.0001), but there was no difference from the group with other causes (p = 0.425). MEN 1 patients were younger than patients with adenoma (p<0.0001), sporadic hyperplasia (p<0.0001), double adenoma (p<0.0001) and other causes (p = 0.019). The adenoma group was also younger than the double adenoma group (p = 0.0227). Preoperative PTH as well as 25OHD values were statistically similar among the groups. Total serum calcium values were lower in the sporadic hyperplasia group than in the

**Table 2. Correlations between acute eGFR changes and preoperative variables.**

| Acute eGFR change (%) vs. | eGFR Preop | Phosphorus Preop | Age | PTH Preop | Total Calcium Preop | Ionized Calcium Preop |
|---|---|---|---|---|---|---|
| n | 494 | 480 | 494 | 494 | 494 | 494 |
| Spearman r | 0.213 | 0.159 | -0.196 | -0.235 | -0.158 | -0.181 |
| 95% CI | 0.124 to 0.298 | 0.068 to 0.248 | -0.282 to -0.107 | -0.319 to -0.148 | -0.245 to -0.068 | -0.267 to -0.091 |
| p | <0.0001 | 0.0004 | <0.0001 | <0.0001 | 0.0004 | <0.0001 |

**Table 3. Demographic characteristics and serum biochemical variables stratified by cause of primary hyperparathyroidism.**

|  | Adenoma | MEN1 | Sporadic Hyperplasia | Double Adenoma | Other Causes [a] |
|---|---|---|---|---|---|
| N [b] | 351 (71.1) | 91 (18.4) | 25 (5.1) | 20 (4.0) | 7 (1.4) |
| Female [b] | 293 (83.5) | 56 (61.5) | 22 (88.0) | 16 (80.0) | 4 (57.1) |
| Non-African American [b] | 293 (83.5) | 82 (90.1) | 21 (84.0) | 20 (100.0) | 6 (85.7) |
| Age (years)[c] | 60 (53–68) | 44 (31–56) | 67 (53.5–75.5) | 64.5 (59–71.8) | 57 (47–63) |
| Creatinine (mg/dL) [c] | 0.82 (0.68–1.03) | 0.78 (0.64–0.9) | 0.84 (0.75–1.21) | 0.82 (0.73–1.09) | 0.73 (0.64–1.19) |
| eGFR (mL/min/1.73 m$^2$) [c] | 83 (61–98) | 102 (83–119) | 80 (52.5–84.5) | 81.5 (53–87) | 92 (60–108) |
| PTH (pg/mL) [c] | 149 (106–254) | 171 (108–224) | 152 (96–196) | 158.5 (119–280) | 226 (128–242) |
| Total calcium (mg/dL) [c] | 10.9 (10.3–11.6) | 10.9 (10.5–11.5) | 10.5 (9.8–11.1) | 10.75 (10.2–12) | 11.1 (9.9–11.9) |
| Ionized calcium (mg/dL) [c] | 5.92 (5.64–6.4) | 6.0 (5.7–6.4) | 5.9 (5.5–6.23) | 5.70 (5.42–6.72) | 6.17 (5.7–6.4) |
| Phosphorus (mg/dL) [c] | 2.7 (2.37–3.1) | 2.6 (2.2–2.9) | 2.9 (2.7–3.3) | 2.8 (2.3–3.2) | 2.6 (2.2–3.9) |
| 25OH vitamin D (ng/mL) [c] | 19 (14–24) | 19 (15–27) | 18.65 (13–22.2) | 20.3 (16–23) | 16 (11–26) |

[a]"Other causes" includes patients with carcinoma (n = 3), MEN 2A (n = 2) and lithium-associated PHPT (n = 2)

[b] Data presented as absolute frequency (percentage)

[c] Data presented as median (first and third quartiles)

adenoma (p = 0.0343) and MEN 1 (p = 0.0218) groups. Ionized calcium showed no difference among the groups.

Oncological follow-up of the three patients treated for parathyroid carcinoma showed cure of cancer in two of them, while the third had lung metastasis with controlled PTH and calcium levels.

Fig 4 shows the mean eGFR in each group stratified by the cause of PHPT. There was a significant reduction in the eGFR from baseline to PO (acute) in all 4 major groups (adenoma,

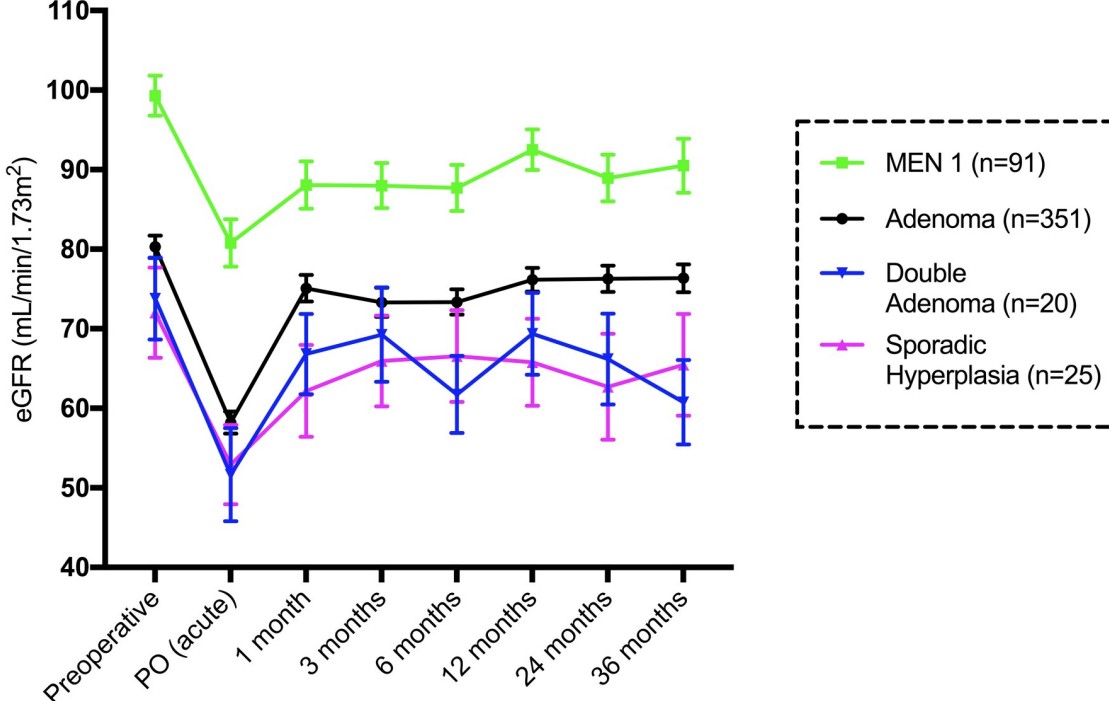

**Fig 4. Mean and standard error of the mean eGFR (mL/min/1.73 m$^2$) until 36 months in patients stratified by the cause of PHPT.** The group with other causes (n = 7) was not included.

MEN 1, double adenoma and sporadic hyperplasia). Adenoma patients had an acute eGFR reduction of -25 mL/min/1.73 m$^2$ (p<0.0001), and all groups had some eGFR recovery at one month.

### Investigation by group according to the AKI classification

Overall, 241 (48.8%) patients met the AKI criteria. Among patients who did not meet the AKI criteria, in 30 (6.1%), serum creatinine decreased in the acute period. Despite the fact that none of the patients fulfilled the criteria for acute injury in the no AKI group, there was a significant difference between the basal eGFR and acute eGFR (p<0.0001). The median acute eGFR reduction was -18 mL/min/1.73 m$^2$, and the preoperative eGFR was higher than the eGFR at 1 month (p = 0.0112). However, there were no differences between the basal eGFR and the eGFR at 12 (p = 0.149) and 24 months (p = 0.106).

Patients with stage 1 AKI (n = 203) had a median acute change in the eGFR of -31 mL/min/1.73 m$^2$, with recovery of +25 mL/min/1.73 m$^2$ at 1 month. The preoperative median eGFR was higher at all time points until 60 months. The eGFR PO was lower than all other eGFR medians (p<0.0001). The baseline eGFR differed by -8 mL/min/1.73 m$^2$ at 1 month (p = 0.0108) (Fig 5).

Subjects with stage 2 AKI (n = 29) had a median acute change in the eGFR of -45 mL/min/1.73 m$^2$, with recovery of +30 mL/min/1.73 m$^2$ at 1 month. The preoperative median eGFR was higher than the eGFR at 12 (p = 0.0319) and 24 months (p = 0.0336). The eGFR PO was lower than all median eGFR values (p<0.0001). The baseline eGFR and the eGFR at 1 month differed by -16 mL/min/1.73 m$^2$ (p = 0.0662). There was no difference between the eGFR at 1 month and other months. In stage 3 AKI (n = 9), the eGFR PO was lower than the basal eGFR PO (p = 0.0003) (Fig 5).

Patients without AKI had a superior basal eGFR to stage 1 AKI (p<0.0001) and stage 3 AKI (p = 0.0004) patients. Subjects without AKI were younger than those with stage 1 (p = 0.0001)

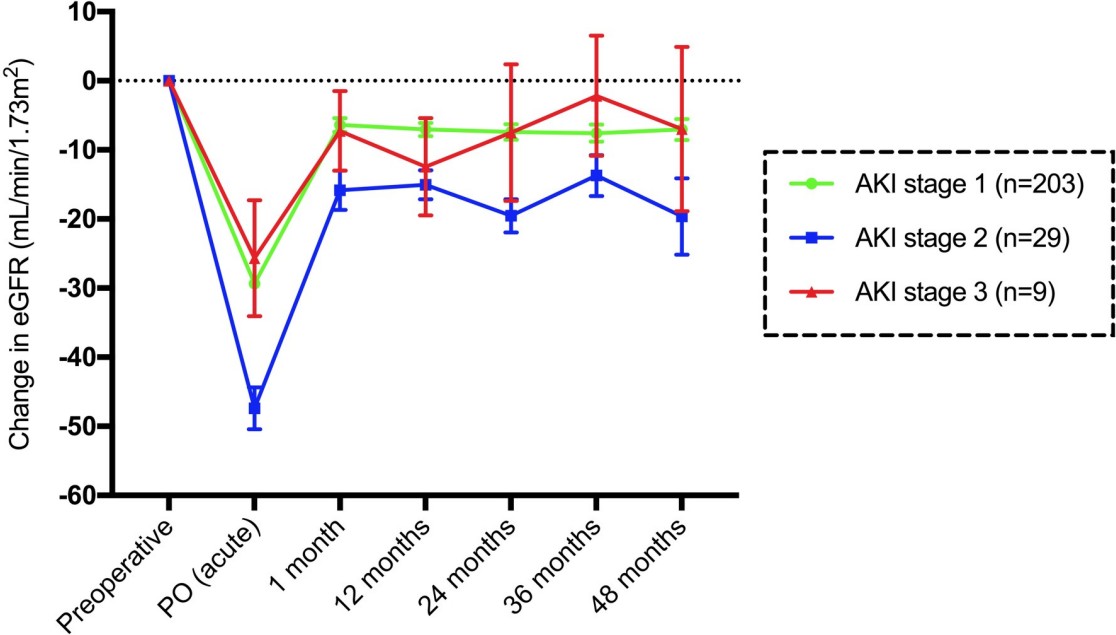

**Fig 5. Mean and standard error of the mean change in the eGFR (mL/min/1.73 m$^2$) compared to baseline until 48 months in patients with AKI divided according to stage.**

**Table 4. Age and preoperative biochemical variables in patients with primary hyperparathyroidism stratified by group according to the AKI classification.**

| | Without AKI n = 253 | Stage 1 AKI n = 203 | Stage 2 AKI n = 29 | Stage 3 AKI n = 9 |
|---|---|---|---|---|
| Age (years) | 56 (44.5–65) [a,b] | 62 (53–68) | 65 (45–76.5) | 57 (51.5–60) |
| Creatinine (mg/dL) | 0.78 (0.65–0.92) [a,c] | 0.86 (0.72–1.17) [d] | 0.76 (0.69–1.08) [e] | 2.20 (1.50–2.72) |
| eGFR (mL/min/1.73 m$^2$) | 94 (74.5–106) [f,g] | 80 (56–96) | 78 (56.5–102.5) | 25 (20.5–64.5) |
| PTH (pg/mL) | 133 (96–191.5) [f,h,i] | 183 (117–298) [j] | 215 (159–349.5) | 1038 (244–1397) |
| Total calcium (mg/dL) | 10.7 (10.2–11.3) [k] | 11 (10.3–11.8) | 11 (10.45–12.3) | 10.9 (10.1–12.7) |
| Ionized calcium (mg/dL) | 5.85 (5.4–6.2) [l,m] | 6.07 (5.7–6.6) | 6.27 (5.82–6.75) | 6.4 (5.7–6.8) |
| 25OH vitamin D (ng/mL) | 19.3 (15–24.9) [n] | 19 (14–24) [o] | 16.3 (9.6–26) | 11.6 (9.5–13.5) |

Data are presented as the median (interquartile range)

[a] p = 0.0001 (versus stage 1 AKI),

[b] p = 0.04 (versus stage 2 AKI),

[c] p = 0.0002 (versus stage 3 AKI),

[d] p = 0.02 (versus stage 3 AKI),

[e] p = 0.005 (versus stage 3 AKI),

[f] p<0.0001 (versus stage 1 AKI),

[g] p = 0.0004 (versus stage 3 AKI),

[h] p = 0.0003 (versus stage 2 AKI),

[i] p<0.0001 (versus stage 3 AKI),

[j] p = 0.01 (versus stage 3 AKI),

[k] p = 0.01 (versus stage 1 AKI),

[l] p = 0.004 (versus stage 1 AKI),

[m] p = 0.02 (versus stage 2 AKI),

[n] p = 0.02 (versus stage 3 AKI),

[o] p = 0.03 (versus stage 3 AKI)

and stage 2 (p = 0.04) AKI. Patients without AKI had significantly lower PTH levels than patients in all three AKI groups. There was also a difference between AKI stage 1 and AKI stage 3 (p = 0.01) patients (Table 4).

Similarly, patients without AKI had lower total calcium levels than those with stage 1 AKI (p = 0.01), but these levels were not lower than those in the other groups. Regarding preoperative 25-hydroxivitamin D, stage 3 AKI patients had an inferior median to those without AKI (p = 0.02) and with stage 1 AKI (p = 0.03) (Table 4).

## Relationship between the change in eGFR and classification of CKD

Preoperatively, 223 (45.1%) patients had an eGFR $\geq$ 90 mL/min/1.73 m$^2$ (G1); 169 (34.2%) had an eGFR of 89–60 mL/min/1.73 m$^2$ (G2); 84 (17.0%) had an eGFR of 59–30 mL/min/1.73 m$^2$ (G3); and 18 (3.6%) had an eGFR of 29–15 mL/min/1.73 m$^2$ (G4).

Using an eGFR of 60 mL/min/1.73 m$^2$ (the cutoff value for an indication of parathyroidectomy in asymptomatic patients), patients were divided into two groups. In 392 (79.3%) patients with a preoperative eGFR $\geq$ 60 mL/min/1.73 m$^2$ (G1 +G2), the median acute change in the eGFR was -24 mL/min/1.73 m$^2$ (p<0.0001), with a recovery compared to baseline of +19 mL/min/1.73 m$^2$ at 1 month (p<0.0001) (Fig 6). After 12 months postoperation, the median permanent change in the eGFR was -5 mL/min/1.73 m$^2$ (p<0.0001), which was sustained over 60 months of follow-up.

The group of 102 (20.6%) patients with an eGFR < 60 mL/min/1.73 m$^2$ (G3 + G4) had a median acute eGFR change of -13 mL/min/1.73 m$^2$ (p<0.0001), which returned to the baseline median at 1 month (p = 0.3076) (Fig 6). After 1 month, the median eGFR was similar to the

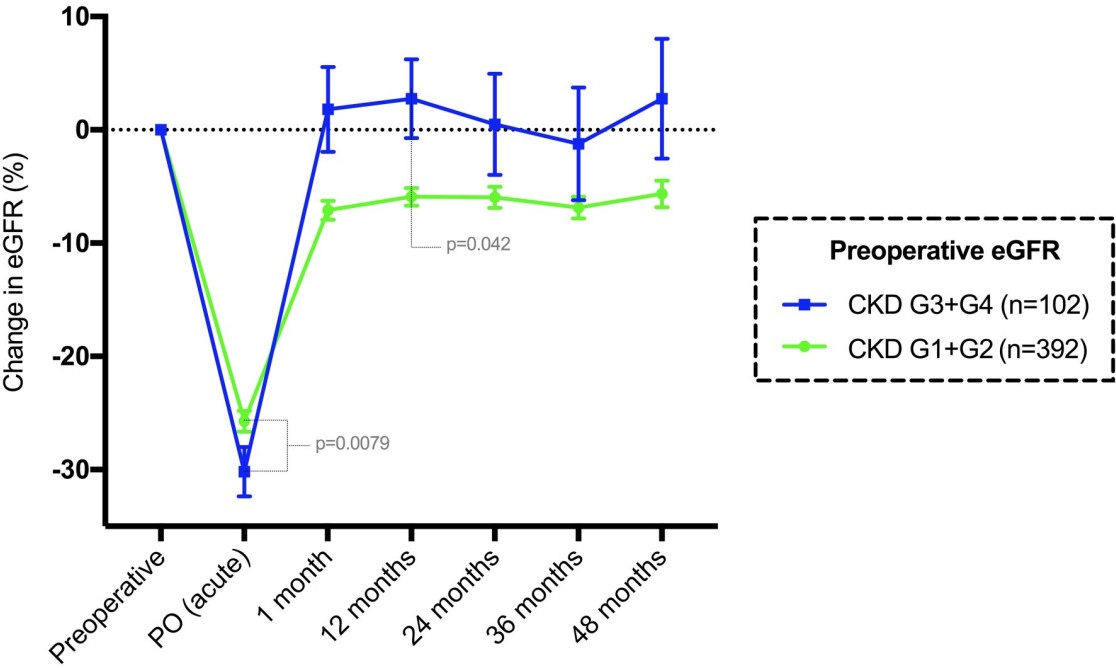

**Fig 6. Mean and standard error of the mean change in eGFR (%) compared to baseline until 48 months in patients classified by CKD grade.** G1+G2 represents a preoperative eGFR $\geq$ 60 mL/min/1.73 m², while G3+G4 represents an eGFR < 60 mL/min/1.73 m².

long-term medians until 60 months. There were significant differences in age and preoperative biochemical data between the two groups (Table 5).

## Postoperative calcium changes over 72 hours

On day 1, there was a median reduction of 1.8 mg/dL (p<0.0001) in total calcium, and the median reduction was 0.3 mg/dL between day 1 and day 2 (p<0.0001). Between days 2 and 3, there was no difference (p = 0.6568). Patients operated on for MEN 1 (most undergoing total parathyroidectomy with autograft), with complete measurements until day 3 (n = 86), showed

**Table 5. Age and preoperative biochemical variables in patients with primary hyperparathyroidism stratified by the preoperative CKD classification.**

|  | G1 + G2 [a] n = 392 | G3 + G4 [b] n = 192 | p value [c] |
|---|---|---|---|
| Age (years) | 56 (46.2 to 66) | 66 (57 to 73) | <0.0001 |
| Cr preoperatively (mg/dL) | 0.75 (0.64 to 0.87) | 1.35 (1.09 to 1.66) | <0.0001 |
| Baseline eGFR (mL/min/1.73 m²) | 93 (80 to 106.8) | 46 (33 to 53.2) | <0.0001 |
| Acute (PO) change in eGFR (%) | -25 (-38.3 to -12.2) | -31.9 (-45.3 to -18.1) | 0.0079 |
| Chronic (12 m) change in eGFR (%) | -5.1 (-14.3 to 3.09) | 0 (-17.9 to 18.35) | 0.0420 |
| PTH preoperatively (pg/mL) | 143 (104 to 218) | 212.5 (125 to 443.8) | <0.0001 |
| CaT preoperatively (mg/dL) | 10.9 (10.3 to 11.5) | 10.9 (10.3 to 11.7) | 0.5522 |
| CaI preoperatively (mg/dL) | 5.91 (5.6 to 6.3) | 6.02 (5.7 to 6.56) | 0.0261 |

[a] G1+G2: eGFR $\geq$ 60 mL/min/1.73 m²

[b] G3+G4: eGFR <60 mL/min/1.73 m²

[c] Data presented as the median (interquartile range)—Mann-Whitney test

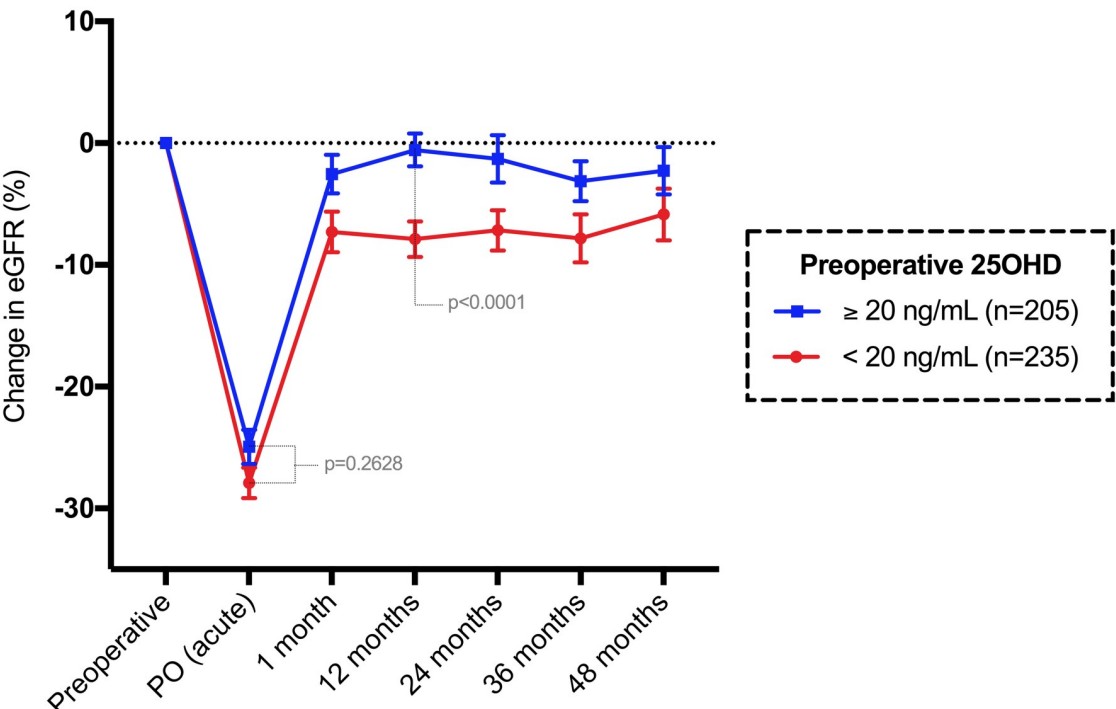

**Fig 7. Mean and standard error of the mean change in eGFR (%) compared to baseline until 48 months in patients classified by 25-OH vitamin D level.**

a further decrease in total calcium until day 3 (p = 0.0001). Ionized calcium showed similar results.

The acute change (%) in total calcium and ionized calcium was correlated with the acute change (%) in eGFR (Spearman $r$ = -0.1511/95% CI -0.2403 to -0.0594/p = 0.0009 for total calcium and Spearman $r$ = -0.1243/95% CI -0.2138 to -0.0328/p = 0.0062 for the change in ionized calcium).

## Relationship between the change in eGFR and preoperative 25-OH vitamin D

Among the 440 patients with preoperative 25-OH vitamin D data, there were 235 patients with values under 20 ng/mL and 205 patients with values ≥ 20 ng/mL. The median (Q1-Q3) preoperative 25OHD value was 19 (14–24) ng/mL. The acute eGFR change (%) was similar among patients (p = 0.2628), using 20 ng/mL as the cutoff value (Fig 7). Conversely, the chronic eGFR change (%) was significantly different among groups (p<0.0001), where patients with 25OHD < 20 ng/dL had a median 6.77% permanent loss in the eGFR after 12 months compared to baseline.

## Multivariate analysis

Table 6 shows that age, preoperative ionized calcium, phosphorus and creatinine influenced the acute change in the eGFR but explained only 13.5% of this percent change ($R^2$ = 0.135). The relative eGFR change at 12 months was explained by the acute (PO) change in eGFR (%), preoperative creatinine, sex and preoperative 25OHD level, explaining 31% of this change.

**Table 6. Multivariate analysis according to model.**

| Model | Factor | Coefficient | Standard error | t value | p | $R^2$ |
|---|---|---|---|---|---|---|
| Acute change eGFR (%) | Constant | -15.26 | 11.76 | -1.30 | 0,195 | 0.135 |
| | Ionized Calcium | 4.99 | 1.44 | 3.47 | 0.001 | |
| | Age | 0.30 | 0.07 | 4.64 | <0.001 | |
| | Phosphorus | -4.46 | 1.63 | -2.74 | 0.007 | |
| | Serum creatinine | 6.54 | 2.37 | 2.77 | 0.006 | |
| Chronic change eGFR (%) | Constant | 9.43 | 3.44 | 2.74 | 0.006 | 0.310 |
| | Acute variation eGFR (%) | 0.53 | 0.05 | 10.64 | <0.001 | |
| | Serum creatinine | -15.08 | 2.39 | -6.30 | <0.001 | |
| | Male (gender) | 7.62 | 2.31 | 3.31 | 0.001 | |
| | 25OHD | -0.34 | 0.11 | -3.10 | 0.002 | |

The acute and chronic changes in the eGFR could be predicted by the following formulae:

$$\text{Acute (PO) change in eGFR } (\%) = -15.26 + 4.99 \text{ x } [\text{preoperative ionized calcium}] + 0.30 \text{ x age}$$
$$-4.46 \text{ x } [\text{preoperative phosphorus}] + 6.54 \text{ x preoperative creatinine}$$

$$\text{Chronic (12 m) change in eGFR } (\%) = 9.43 + 0.53 \text{x } [\text{acute eGFR change } (\%)]$$
$$- 15.08 \text{ x } [\text{preoperative creatinine}] + 7.62 \text{ (if male)} - 0.34 \text{ x } [25\text{OHD}]$$

## Discussion

The present study shows that a significant number of patients subjected to curative parathyroidectomy for PHPT apparently suffer from acute deterioration of renal function, despite the type and extent of the procedure. We observed that 92.7% of patients had some reduction in the eGFR during the first 4 days after surgery; of those, more than half fulfilled the KDIGO criteria for AKI. It is remarkable that in most patients, the rise in serum creatinine was not accompanied by symptoms, and only one patient in the present series needed dialysis during the hospital stay (0.2%). This non-oliguric silent impairment of renal function is the probable explanation for the lack of studies on this issue.

The cohort analyzed here included a majority of symptomatic patients. As far as we are concerned, data regarding renal changes in asymptomatic patients after parathyroidectomy are lacking. The change in glomerular filtration may be more pronounced and clinically evident in advanced disease, as noted in the past [22].

Despite the high incidence of AKI (48.8%), fortunately, most patients had stage 1 AKI (41.1%), while the other 7.7% had stage 2 and 3 AKI. The frequency of postoperative AKI has been shown to be lower in general surgery, and even among high-risk cardiac surgery, it can reach rates of 10 to 22% [23–25]. In contrast to our results, otolaryngology procedures, including neck surgeries, have an overall AKI incidence of only 4.1% [26].

Fortunately, we did not observe an increase in hospital mortality in our patients. The mortality rate has been shown to correlate with AKI stage in a progressive way, despite the many classifications used in past years [27–29]. In addition to the higher mortality in AKI patients, there is also evidence of an increase in morbidity and risk of progression to CKD [25, 30].

A prospective study of 62 patients treated surgically for PHPT reported a small but significant eGFR decrease of -3 mL/min/1.73 m$^2$ acutely and -5 mL/min/1.73 m$^2$ 60 days after parathyroidectomy, but they considered preoperative comorbidities as the cause of renal

dysfunction [10]. In that study, creatinine was measured only on day 1 after surgery. Due to our health system and social conditions (symptomatic, low-income patients residing far from the hospital), our postoperative routine since the 1990s has been to keep patients in the hospital for at least 3 days, with the aim of observing and correcting possible hypocalcemia, with biochemical assessment on a daily basis. We observed in our study that the highest creatinine value was found on day 1 in only 44.5% of patients, while it was found from days 2 to 4 in the remaining 55.5%. Thus, early discharge is probably a factor in masking acute kidney dysfunction. This study is the first to show those curves in detail, with a robust cohort of patients. Long-term data were collected until 5 years of follow-up and confirmed a small but permanent reduction in kidney function.

In this cohort, patients with less pronounced acute eGFR impairment tended to be younger, have lower PTH and calcium levels, and have better preoperative kidney function. Conversely, those who met the criteria for stage 3 AKI had higher PTH levels, were older and had a worse preoperative eGFR. We also identified that an acute change in the eGFR (%) was statistically correlated with advanced age, higher PTH, and total and ionized calcium and phosphorus levels, as well as elevated preoperative creatinine and lower basal eGFR values in univariate analysis. In multivariate analysis, age and preoperative ionized calcium, phosphorus and creatinine levels were variables associated with acute change.

Clinically, the hyperactivity of the parathyroid glands is associated with the functional effect of GFR elevation [31]. Although the present data are not supportive of hyperfiltration in PHPT (not in strict definition), some attention should be paid to the possible overestimation of the eGFR preoperatively due to PHPT. The measured clearance of inulin was 100% higher in patients with hypercalcemia due to PHPT than in patients with PTH-independent hypercalcemia, controlling for the level of calcium [32, 33]. The consequence of this hypothesis of glomerular function overestimation in some cases of PHPT is that the GFR threshold of 60 mL/min/1.73 m$^2$ in the criteria for surgery in asymptomatic PHPT may postpone definitive treatment in some patients whose renal function is actually worse than is apparent from current biochemical evaluations [12].

In the literature, the results of long-term renal function after surgical treatment of PHPT are conflicting. Some authors believe that parathyroidectomy is associated with a further decrease in the eGFR [34]. Our results contrast those of a prospective study that did not observe significant differences in renal function preoperatively or 12 months after the operation in 19 patients [35]. The small sample size and preoperative kidney function may be factors of these conflicting results.

It has been suggested that long-term kidney function does not deteriorate after parathyroidectomy unless complicating factors (such as urinary infection or hypertension) are present [36]. However, when renal function reserve is lower, parathyroidectomy may have an impact, similar to renal transplant patients after parathyroidectomy [37].

Some retrospective studies addressing long-term renal changes after parathyroidectomy for PHPT showed that patients with a higher preoperative eGFR had more permanent kidney function after parathyroidectomy (PTX) in contrast to patients with a lower preoperative eGFR, suggesting that parathyroidectomy alone is able to halt the deterioration of renal function in those patients with presurgical KDIGO grade 3 CKD or higher [11, 38]. In our opinion, this conclusion underestimates the role of acute and permanent renal impairment after PTX, which is clearly shown in our results. Other authors have small and selected case series suggesting that parathyroidectomy slows the decline in kidney function but demonstrating an almost significant long-term decline in the eGFR [39].

A small series showed that parathyroidectomy caused loss of renal function not observed in patients treated with denosumab. The preoperative eGFR values in both groups were not

comparable and were higher in the parathyroidectomy group [40]. In our experience using CKD grade, we found a median permanent reduction in the eGFR 1 month after parathyroidectomy in 392 patients with G1+G2 CKD (preoperative eGFR ≥ 60 mL/min). Conversely, in 102 patients with G3+G4 CKD (preoperative eGFR < 60 mL/min), the median reduction in the eGFR was not significant. Thus, the conclusion that parathyroidectomy is worse than medical management with denosumab should be carefully reviewed. Indeed, in prospectively compared asymptomatic patients, there were no significant differences in creatinine levels up to two years of medical follow-up or surgery [41].

One important aspect is the impact of these changes compared to basal changes. In Tassone's study [11], the reduction observed in group 1 was 5.2 mL/min in 86.8 mL/min or 6%. The theoretical reduction in group 2 theoretically was 2.4 ml/min of the baseline of 52.6 mL/min or almost 5%. Apparently, both groups had the same amount of reduction. Independent of long-term renal function, both groups had an acute reduction in the eGFR in the first 96 h of the postoperative period, which has not been fully explored in other studies.

The results of eGFR postoperative patterns in patients stratified by diagnosis, and hence the extent of surgery, confirmed the same pattern of acute renal impairment, independent of the cause of PHPT. Patients with MEN1-related PHPT were younger and had better kidney function prior to surgery than patients in the other groups. The expressive number of MEN1 PHPT in the present series is a consequence of a very specialized endocrinological unit dedicated to genetic diseases, leading to an increasing number of parathyroid operations [42]. Basal PTH as well as preoperative 25OHD values were statistically similar among groups. Total serum calcium levels were slightly lower in the sporadic hyperplasia group than in the adenoma and MEN 1 groups. These facts may suggest that the degree of PTH reduction is more important than the other aspects in correlation to the acute increase in creatinine. We could not test this hypothesis in our cohort because early postoperative PTH sampling was not routine in our practice. Whether the fall in intraoperative PTH sampling could be employed as a surrogate marker of this acute decrease in the eGFR may be explored in future studies.

The type and mechanism of kidney damage caused by PTX is not well understood and cannot be answered by this study. The acute impact caused by parathyroidectomy seems to be out of proportion with operative time and surgical trauma, as also compared previously in tertiary HPT [4]. We showed that patients treated for adenomas, typically with a short localized surgery, presented with the same pattern of kidney dysfunction as those with more extensive procedures, such as total parathyroidectomy with auto transplantation performed in MEN 1-related PHPT. We previously observed that renal dysfunction was not observed in patients undergoing other head and neck procedures [43].

Chronic hypercalcemia is a well-recognized cause of kidney dysfunction, and excessively high serum calcium levels indicate surgical treatment in asymptomatic PHPT [12]. In addition to structural changes due to intratubular plugging, peritubular interstitial inflammation and fibrosis, there is a direct vasoconstrictive effect of calcium on afferent arterioles [44, 45]. We observed a weak but significant positive correlation of the percent acute variation in calcium and acute variation in the eGFR.

There is recent evidence that 25-OH vitamin D depletion (deficient <20 ng/mL) in PHPT is associated with increased severity, reflected in biochemical and bone markers [46]. We were not able to show a correlation between preoperative 25OHD levels and acute eGFR variations. Our median preoperative 25OHD level was 19 ng/mL, which was considered lower than that in other PHPT studies, and could possibly impact the long-term recovery of kidney function after an AKI episode, as demonstrated in an ischemia/reperfusion model in rats [47, 48]. The striking difference in eGFR recovery when patients were stratified by the observed cutoff value of 20 ng/mL for preoperative 25-OHD levels deserves further investigation. There is some

concern regarding supplementing vitamin D in HPTP patients before parathyroidectomy and increasing calcium levels. Considering that this calcium increase is apparently low and that a possible protective effect may ensue, vitamin D supplements should be given to patients waiting for surgery until proven otherwise [49].

Kidney injury is a major health problem, and there are many severe complications of CKD, compromising quality of life and survival. Recently, the International Society of Nephrology and the International Federation of Kidney Foundations issued an alert with strong recommendations regarding the awareness of patients at risk of developing AKI [50]. The data presented here warrant a policy of creatinine surveillance before and after parathyroidectomy, as well as avoidance of any nephrotoxic drug or dehydration in these patients. Although most patients will demonstrate minor changes, fortunately with minimal clinical impact, some may have significant morbidity, even young patients undergoing parathyroidectomy [51].

If one considers the global concern of preventing global epidemics of CKD, the present data raise further attention to renal function in PHPT [52]. A better understanding of mechanisms will probably contribute to improving patient care, as parathyroidectomy has many benefits in most cases [53, 54]. In the future, biomarkers of renal cell injury may help to identify additional patients with AKI at an earlier stage [19].

The present study has several limitations. It is retrospective, and we have no data on comorbidities, body mass changes, or medications that may affect renal function. There is not a control group of patients with PHPT who were not subjected to parathyroidectomy in order to clarify what would be the natural course of kidney function in untreated patients. Ethical reasons may preclude a prospective trial like this in symptomatic cases.

## Supporting information

**S1 File.**
(XLSX)

## Author Contributions

**Conceptualization:** Marcelo Belli, Munro Peacock, Fábio Luiz de Menezes Montenegro.

**Data curation:** Marcelo Belli, Climério Pereira Nascimento, Jr, Fábio Luiz de Menezes Montenegro.

**Formal analysis:** Marcelo Belli, Munro Peacock, Fábio Luiz de Menezes Montenegro.

**Investigation:** Marcelo Belli, Fábio Luiz de Menezes Montenegro.

**Methodology:** Marcelo Belli, Regina Matsunaga Martin, Marília D'Elboux Guimarães Brescia, Fábio Luiz de Menezes Montenegro.

**Project administration:** Marcelo Belli, Fábio Luiz de Menezes Montenegro.

**Software:** Marcelo Belli, Fábio Luiz de Menezes Montenegro.

**Supervision:** Bruno Ferraz-de-Souza, Rosa Maria Affonso Moyses, Munro Peacock, Fábio Luiz de Menezes Montenegro.

**Validation:** Bruno Ferraz-de-Souza, Rosa Maria Affonso Moyses, Munro Peacock, Fábio Luiz de Menezes Montenegro.

**Visualization:** Marcelo Belli, Regina Matsunaga Martin, Marília D'Elboux Guimarães Brescia, Climério Pereira Nascimento, Jr, Ledo Mazzei Massoni Neto, Sergio Samir Arap, Bruno

Ferraz-de-Souza, Rosa Maria Affonso Moyses, Munro Peacock, Fábio Luiz de Menezes Montenegro.

**Writing – original draft:** Marcelo Belli, Munro Peacock, Fábio Luiz de Menezes Montenegro.

**Writing – review & editing:** Marcelo Belli, Munro Peacock, Fábio Luiz de Menezes Montenegro.

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
