## [Decision Letter · Decision Letter 0]

7 Sep 2020

PONE-D-20-22587

Acute and long-term kidney function after parathyroidectomy for primary hyperparathyroidism

PLOS ONE

Dear Dr. Belli,

Thank you for submitting your manuscript to PLOS ONE. After careful consideration, we feel that it has merit but does not fully meet PLOS ONE’s publication criteria as it currently stands. Therefore, we invite you to submit a revised version of the manuscript that addresses the points raised during the review process.

We look forward to receiving your revised manuscript.

Kind regards,

Itamar Ashkenazi

Academic Editor

PLOS ONE

Journal Requirements:

2. In ethics statement in the manuscript and in the online submission form, please provide additional information about the patient records used in your retrospective study. Specifically, please ensure that you have discussed whether all data were fully anonymized before you accessed them and/or whether the IRB or ethics committee waived the requirement for informed consent. If patients provided informed written consent to have data from their medical records used in research, please include this information.

Reviewers' comments:

Reviewer's Responses to Questions

**Comments to the Author**

1. Is the manuscript technically sound, and do the data support the conclusions?

Reviewer #1: Yes

Reviewer #2: Partly

Reviewer #3: Yes

2. Has the statistical analysis been performed appropriately and rigorously? 

Reviewer #1: Yes

Reviewer #2: No

Reviewer #3: Yes

3. Have the authors made all data underlying the findings in their manuscript fully available?

Reviewer #1: Yes

Reviewer #2: Yes

Reviewer #3: Yes

4. Is the manuscript presented in an intelligible fashion and written in standard English?

Reviewer #1: No

Reviewer #2: No

Reviewer #3: Yes

5. Review Comments to the Author

Reviewer #1: Well written article regarding a large retrospective cohort of patients operated for PHPT showing in an important percentage various degrees of kidney dysfunction, mostly acutely, but some with long term persistance, suggesting that more cation should be invested in their renal follow-up.

Authors do not mention which was the time span of hyperparathyroidism before surgery.

It is also not clear how come so many patients were having MEN1, because he ratio is way above literature data. They should comment on that.

Table 1 – authors should underline significant changes compared to baseline at different time points (e.g. p values)

Table 2 – the information regarding p values needs to be simplified, e.g. * compared to AKI stage 1, # compared to AKI stage 2, $ compared to AKI stage 3. One sign – p < 0.05, two signs – p < 0.01, three signs – p < 0.001.

Patients with AKI type 3 had clearly higher PTH levels, but also lower 25OH-D3, authors should comment more in detail on that (they already commented in discussion)

Authors should make discussions more concentrated and more focused on the main message - that parathyroidectomized patients should be more attentively followed regarding kidney function. They should add separate comments on study limits - retrospective study, small control group, etc. - and conclusions - follow up of kidney function in this type of patients, the effect of surgery itself with decrease of PTH on kidney function, which may be important also in secondary hyperparathyroidism, the importance to revisit vitamin D repletion before surgery in PHPT.

Finally, English needs to be improved

Introduction

- after the acute episode: during the acute period

- patients who underwent parathyroidectomy

- during the first month after surgery

- patients operated for PHPT seem to present…. When compared to control thyroidectomized patients

Main text

- row 56 – Other authors found that, despite…..most patients recover

- row 86 – with a control group

- row 496 – PHPT

- row 549 – this difference disappeared however at 24 months

- row 559 the authors concluded that (without comma)

- row 607 – dysfunction, not disfunction

etc.

Reviewer #2: The aim of this retrospective study, was to investigate both the acute and chronic effects of parathyroidectomy on renal function and relate these to serum biochemistry before surgery in 494 patients with surgically proven PHPT, and to compare with control group.

The authors found that there is a significant acute impairment in renal function after parathyroidectomy for PHPT and almost half of patients meet the criteria for AKI. Significant eGFR recovery was observed during first month after surgery, but a small permanent reduction may occur.

Comments

1. The discussion is too long. Please condense it.

2. Please show serial changes in body weight over time.

3. You mentioned that permanent reduction in eGFR occurred in 60.7 % of the patients, after acute episode. Please reveal independent factors contributing to permanent reduction in eGFR by multivariate analysis.

4. Did you look at blood urea nitrogen (BUN) or uric acid (UA) and their serial changes?

5. Table 6 showed a significant difference in preoperative GFR between PHPT and control. This is a major limitation. Please describe this as a limitation of this study in the main text.

6. How do you feel about an association of acute kidney impairment after surgery with renal tubular damage? Or did you look at biomarkers of renal tubular damage, such as beta2 microglobulin or liver-type fatty acid protein?

Reviewer #3: I commend Belli and col for this interesting article which evaluates changes in renal function following parathyroidectomy for PHPT. The conclusion is interesting and may have important practical conséquences. One major strengths of the article is the long in hospital stay and ability to evaluate renal function several days after PHPT. As in most centers PHPT patients are currently operated on an outpatient basis the present study design would be difficult to reproduce.

I have several comments.

1. The authors have included PHPT for adenoma, hyperplasia, MEN1 and parathyroid cancer. Although all these patients have primary hypeparathytoidism these are different diseases with different mechanisms of PHPTand diffrent treatments...I think the article should focus only on patients with PHPT related to parathyroid adenoma which is the most usual cause of PHPT

2. I Don't see the utility of the control group. Thyroidectomy can result in ablation of one parathyroid gland without significant impact on the overall parathyroid function.

3. The result section is much too long. as a result it is quite difficult to read.I think that the chapters on relationships between eGFR and AKI, CKD and vit D should only be cited and presented in Tables. Tables and Figures should be counted separately and some should be published as suplementarry material because there are too many of them…

4. I think an uni/multivariate analysis would be helpful in identifying patients at risk of developping renal disfunction after parathyroidectomy for PHPT.

5. How many patients developped clinically relevant renal failure? How were they managed? Could the authors identify risk factors for such complication?

6. The discussion is much too long. the authors should stick to their results rather than discuss tertiary parathyroidism and chronic kidney disease....2 or three pages are enough in my opinion

6. PLOS authors have the option to publish the peer review history of their article (what does this mean?). If published, this will include your full peer review and any attached files.

Reviewer #1: No

Reviewer #2: No

Reviewer #3: No

---

## [Author Response · Author response to Decision Letter 0]

17 Nov 2020

We appreciate the opportunity to revise and resubmit the manuscript.

We have worked to run multivariate analysis, to short data presentation and discussion, and to review professionally the manuscript.

---

## [Decision Letter · Decision Letter 1]

4 Dec 2020

Acute and long-term kidney function after parathyroidectomy for primary hyperparathyroidism

PONE-D-20-22587R1

Dear Dr. Belli,

We’re pleased to inform you that your manuscript has been judged scientifically suitable for publication and will be formally accepted for publication once it meets all outstanding technical requirements.

Kind regards,

Itamar Ashkenazi

Academic Editor

PLOS ONE

Additional Editor Comments (optional):

Reviewers' comments:

Reviewer's Responses to Questions

**Comments to the Author**

1. If the authors have adequately addressed your comments raised in a previous round of review and you feel that this manuscript is now acceptable for publication, you may indicate that here to bypass the “Comments to the Author” section, enter your conflict of interest statement in the “Confidential to Editor” section, and submit your "Accept" recommendation.

Reviewer #1: All comments have been addressed

Reviewer #2: (No Response)

Reviewer #3: All comments have been addressed

2. Is the manuscript technically sound, and do the data support the conclusions?

Reviewer #1: Yes

Reviewer #2: (No Response)

Reviewer #3: Yes

3. Has the statistical analysis been performed appropriately and rigorously? 

Reviewer #1: Yes

Reviewer #2: (No Response)

Reviewer #3: Yes

4. Have the authors made all data underlying the findings in their manuscript fully available?

Reviewer #1: Yes

Reviewer #2: (No Response)

Reviewer #3: Yes

5. Is the manuscript presented in an intelligible fashion and written in standard English?

Reviewer #1: Yes

Reviewer #2: (No Response)

Reviewer #3: Yes

6. Review Comments to the Author

Reviewer #1: Well written article regarding a large retrospective cohort of patients operated for PHPT showing in an important percentage various degrees of kidney dysfunction, mostly acutely, but some with long term persistance, suggesting that more caution should be invested in their renal follow-up.

All advices were successfully tackled by the authors and improved paper quality. They detailed in discussions the suggestions made. Also, discussions stressed the importance of kidney function follow-up in these patients, as well as vitamin D measurement. The authors finely tuned their English language as adviced. The changes of tables 1 and 2 are satisfactory.

I believe that the paper in the actual form can be accepted for publication in PLOS One.

Reviewer #2: (No Response)

Reviewer #3: (No Response)

7. PLOS authors have the option to publish the peer review history of their article (what does this mean?). If published, this will include your full peer review and any attached files.

Reviewer #1: No

Reviewer #2: No

Reviewer #3: No

---

## [Editor Report · Acceptance letter]

18 Dec 2020

PONE-D-20-22587R1 

Acute and long-term kidney function after parathyroidectomy for primary hyperparathyroidism 

Dear Dr. Belli:

I'm pleased to inform you that your manuscript has been deemed suitable for publication in PLOS ONE. Congratulations! Your manuscript is now with our production department. 

Kind regards, 

on behalf of

Dr. Itamar Ashkenazi 

Academic Editor

PLOS ONE